# U.S. Fruit and Vegetable Affordability on the Thrifty Food Plan Depends on Purchasing Power and Safety Net Supports

**DOI:** 10.3390/ijerph19052772

**Published:** 2022-02-27

**Authors:** Sabrina K. Young, Hayden Stewart

**Affiliations:** U.S. Department of Agriculture, Economic Research Service, Kansas City, MO 64104, USA; hayden.stewart@usda.gov

**Keywords:** diet quality, fruits and vegetables, socioeconomic status, SNAP, food stamps, safety net, thrifty food plan

## Abstract

The Supplemental Nutrition Assistance Program (SNAP) increases the food purchasing power of lower-income households so that they can better afford a nutritious diet. Benefit amounts are based in part on the cost for a household to follow the Thrifty Food Plan (TFP), a meal pattern designed to meet the Dietary Guidelines for Americans. In October 2021, the U.S. Department of Agriculture (USDA) reformulated the TFP and increased its cost by 21%. However, the TFP still does not account for differences in food prices across the country. This study examines: (1) how geographic variation in food prices affects fruit and vegetable affordability and (2) to what extent raising the cost of the TFP (and therefore the maximum SNAP benefit) has mitigated these effects. We use data on fruit and vegetable prices from the USDA and simulation techniques to create and cost out food baskets with a sufficient quantity and variety of fruits and vegetables for a representative household to satisfy guidelines over one week. We find that the increase in SNAP benefits will increase fruit and vegetable affordability for participating households. However, households facing food prices greater than national average food prices may still face tradeoffs to purchase a balanced diet.

## 1. Introduction 

The United Nations (UN) placed consumption of produce at the forefront in 2021 by designating it as the International Year of Fruits and Vegetables as part of the UN Decade of Action on Nutrition (2016–2025), citing the affordability of fruits and vegetables as an important barrier on which to focus [1]. Underconsumption of these foods is a problem in many countries including European Nations and the United States (U.S.) [2,3,4].

Social safety nets are among the mechanisms available to governments around the world for “pulling” lower-income households closer to fruit and vegetable consumption goals according to Research Partners of the Scientific Group for the Food Systems Summit in March 2021 [5]. In the U.S., the Supplemental Nutrition Assistance Program (SNAP) is uniquely placed in this role. As the largest U.S. food assistance program, SNAP aims “to increase the food purchasing power of lower-income households in order to obtain a more nutritious diet” that satisfies Federal recommendations for fruits, vegetables, and other food groups [6].

To provide U.S. households with the purchasing power they need to eat a healthy diet consistent with Federal recommendations, the U.S. Department of Agriculture (USDA) sets a SNAP household’s benefits based on the Thrifty Food Plan (TFP) which, in turn, is the cost of the groceries for one week of budget-conscious and healthy meals for a four-person family [7,8]. Benefit amounts are adjusted for other household sizes. Households with zero net income receive the full cost of the TFP. Those with positive net income receive a reduced benefit equal to the maximum amount for their household minus 30 percent of their net income. (Net income is calculated as gross income minus allowable deductions such as excess shelter costs and 20 percent of earned income.)

Increasing lower-income households’ level of fruit and vegetable consumption would reduce identified differences in diet quality between them and other households as well as improve overall health and wellbeing. Lower-income individuals consume even fewer servings of fruits and vegetables than others do [4,9]. In 2017–2018, members of households with incomes below 185% of the federal poverty line (FPL) consumed a daily average of 0.88 cups of fruit and 1.24 cups of vegetables, whereas those in households with incomes over 300% of the FPL consumed 0.97 and 1.53 cups, respectively [4]. (Households between 185 and 300% fall in line accordingly.) SNAP participants also score lower than both income-eligible and ineligible nonparticipants for both food groups on the Healthy Eating Index 2020, a measure of conformance with Federal dietary recommendations [10]. Research also shows that households with marginal, low, and very low food security are successively more likely to suffer from diet-related chronic illnesses, such as hypertension and diabetes, compared with their high food secure peers [11]. 

Dietary goals for members of the U.S. population at all income levels are set every 5 years by the USDA and the U.S. Department of Health and Human Services. Each new version of the *Dietary Guidelines for Americans* (*Dietary Guidelines*) sets goals for the quantity and variety of fruits, vegetables, and other foods that should be consumed given an individual’s age, gender, and level of physical activity [12]. In addition to meeting *Dietary Guidelines* food group and subgroup requirements, households require enough variety of fruits and vegetables to cook meals containing the meats, grains, and other types of foods they also eat. To ensure that households’ SNAP benefits are sufficient, USDA creates the TFP using an optimization model [7,8]. Food intake surveys are analyzed to identify what meals and snacks individuals most consume. Food databases with recipes next convert those meals and snacks into ingredients purchasable at retail food stores. Finally, the optimization model identifies a nutritious diet that is as close as possible to prevailing consumption patterns while still meeting a cost threshold. SNAP households should therefore be able to afford the fruits and vegetables used as ingredients in the commonly consumed meals and snacks identified by USDA researchers. This does not mean that SNAP household can afford every type of fruit and vegetable. Indeed, when the TFP was created in 1975, it was assumed that households would rely on lower-cost items. 

Although USDA sets households’ maximum SNAP benefits based on the TFP, SNAP and other lower-income households continue to express the opinion in surveys and focus group analyses that fruits and vegetables are not affordable [13,14,15,16,17]. A 2018 survey of SNAP participants found that 61% viewed the affordability of healthy food as a barrier to the adequacy of SNAP benefits [17]. Moreover, 38% of respondents stated that fresh vegetables were difficult to afford, and 43% responded that fresh fruits were. 

Fruits and vegetables generally are cheaper than energy-dense foods that should only be consumed in moderation and contribute little to meeting dietary recommendations on both a dollars-per-edible grams basis ($/edible grams) and on a dollars-per-portion basis ($/average portion) [18]. The TFP also shows that a healthy diet is attainable with a food budget equal to a household’s maximum SNAP benefits as discussed above. However, a household’s SNAP benefits may not go equally far in all parts of the Nation. The TFP assumes all SNAP participants to face national average prices whereas households’ situations vary. Some pay higher-than-average food prices because they lack access to large grocery stores or reside in parts of the country, such as some metro areas or rural areas, where food prices exceed the national average prices used by USDA while constructing the TFP [19,20,21]. Indeed, food prices can be as low as 85% of the national average in some places and 129% of the national average in others [22]. 

Households that desire to satisfy the *Dietary Guidelines* recommendations on a limited budget must also be willing to prioritize healthy foods like fruits and vegetables to a greater extent than U.S. households generally do. Research shows that a household receiving maximum SNAP benefits and facing national average prices can afford a sufficient quantity and variety of fruits and vegetables with 40% of those benefits [23]. This is consistent with the TFP which also assumes that households allocate about 40% of their food budgets to fruits and vegetables [7,8]. However, it exceeds the levels most households are willing to budget for fruits and vegetables. U.S. households allocate only about 26% of their food dollars to fruits and vegetables, on average [18].

In June 2021, USDA “modernized” the TFP, raising the real value of households’ SNAP benefits for the first time since 1975 [24]. Previous updates to the TFP had been cost neutral. Increases in the cost of the food plan were made only to adjust for inflation. “Too many of our fellow Americans struggle to afford healthy meals,” according to the Department. “The revised plan is one step toward getting them the support they need to feed their families” [24]. Households began receiving the higher SNAP benefit level based on the new TFP, which is 21% greater than the previous TFP, in October 2021.

In this study, we use simulation techniques similar to those in Stewart et al. [23] and USDA data on fruit and vegetables prices to ask how well a representative household receiving maximum SNAP benefits can afford food baskets that contain a sufficient variety and quantity of fruit and vegetables. We construct 1000 food baskets that, when evaluated at national average prices, will cost about 40% of the historic TFP. Given the new, higher level of benefits SNAP households are receiving, participating households may be able to afford these 1000 baskets with a share of their food budgets that is more consistent with societal norms and therefore allow the households to better maintain their spending on other types of foods. SNAP households may even be able to incorporate more variation of fruits and vegetables into their diet than the historic TFP envisioned. However, even with a 21% increase in SNAP benefits, households’ situations may continue to vary with some households still needing to spend 40% or more of their food budgets on fruits and vegetables in order to afford a sufficient quantity and variety of these foods. 

This study did not evaluate whether SNAP households’ food purchasing power was increased; rather we seek to identify where gaps may remain. To this end, our analysis answers two key questions: (1) How much does regional price variation affect fruit and vegetable affordability with a food budget based on the Thrifty Food Plan? (2) Are the effects of regional price variation on fruit and vegetable affordability mitigated by the October 2021 increase in the cost of the TFP and therefore the maximum SNAP benefit? 

## 2. Materials and Methods

Data from the USDA Fruit and Vegetable Prices data product are used for the study [25]. This data product includes national average prices for 157 commonly consumed fresh and processed fruits and vegetables, including the price by cup equivalent, the unit of measurement in which the *Dietary Guidelines* fruit and vegetable recommendations are stated. 

USDA fruit and vegetable cost estimates used in this study are based on 2016 scanner data provided by grocery stores and other food retailers to Information Resources Incorporated (IRI), a market research company [26]. USDA uses these data to estimate national average retail prices for different food products across retail outlets throughout the U.S. as well as by package size, brand name, and season. However, unlike retail prices that capture the cost of items as sold in stores, prices per cup equivalent have been adjusted to include only the weight of edible parts and account for changes in weight that may occur when items are cooked. Fresh potatoes, for example, lose about 19% of their weight when baked in their skins [27]. Raw apples lose about 10% of their weight when the inedible stem and core are removed [27]. In general, a cup equivalent is the amount of edible fruit or vegetable that will fit in a standard 8-ounce measuring cup [25]. Exceptions include raw leafy vegetables, for which a cup equivalent is 2 cups, and raisins and other dried fruits, for which a cap equivalent is one-half cup. 

We use the Consumer Price Index (CPI) to adjust USDA fruit and vegetable prices for food price inflation that occurred between 2016 and 2020. The U.S. Bureau of Labor Statistics (BLS) publishes indexes to measure the average change over time in prices urban consumers pay for market baskets of goods and services, including baskets of fresh and processed fruits and vegetables [28]. Fresh vegetable prices rose 7.6 percent between 2016 and 2020. Fresh fruit prices fell by 0.2 percent and prices for processed fruits and vegetables rose by 2.3 percent over those same years.

Following the *Dietary Guidelines*, we first create 1000 food baskets that are sufficient for a representative household to satisfy fruit and vegetable recommendations over one week. Members of the household are moderately active and include one adult male (31 to 45 years old), one adult female (31 to 45 years old), and two children (one aged 6–8 years old and another aged 9–11 years old). Recommended consumption for this family includes 7 cup equivalents of dark green vegetables, 22 of red/orange vegetables, 6.5 of legumes, 21 of starchy vegetables, and 17 of other vegetables each week. It also needs 49 of cup-equivalents of fruit. To allow for food loss due to spoilage and other factors that can decrease consumption, we added 5% more food in each basket above the minimum (129 cup equivalents in total). The TFP makes a similar assumption of 5% food loss.

To keep the simulated baskets consistent in cost and variety with the historic TFP, which envisioned SNAP households to consume only lower-cost food items, we include in our baskets exclusively products costing less than $0.90 per cup equivalent in 2020. Shown in Table 1 and Table 2 are selected fruits and vegetables available in this cost range. Specifically, we construct the baskets so that half of all items will cost less than $0.45 per cup equivalent and half will cost between $0.45 and $0.90 cents, on average. Defining our first price interval this way allows us to include products from the fruit category and each vegetable subcategory. Defining our second price interval this way makes it possible to include key, top-consumed products. Previous research shows that baskets created in this manner contain a sufficient variety of fruits and vegetables and are affordable at national average prices for about 40% of the historic TFP [23].

Next, using simulation techniques reported in [23] and the Gauss statistical software package version 19, we selected the products to include in each of the 1000 food baskets. The computer code used to perform these simulations is available in the Appendix A. A list of products in the USDA Fruit and Vegetable Prices data product costing less than $0.90 per cup equivalent in 2020 served as the sampling frame. Products were drawn from this frame one-half of a cup equivalent at a time, with replacement, until we had drawn 104 servings of fruit, 14 servings of dark green vegetables, 46 servings of red/orange vegetables, 14 servings of legumes, 44 servings of starchy vegetables, and 36 servings of other vegetables. Juice was limited to 50% or less of all fruit in any basket. We also used sample weights to ensure that 50% of all items would cost less than $0.45 per cup equivalent, on average. Among all products in both price intervals, 30.6% (34 out of 111) cost less than $0.45 per cup equivalent while 69.4% (77 out of 111) cost between $0.45 and $0.90 per cup equivalent. We therefore set sample weights at 1.63 (0.5/0.306) for items in the first price interval and 0.72 (0.5/0.694) for those in the second price interval. 

After creating the 1000 baskets, we next estimated each of their total costs at four different price levels. USDA fruit and vegetable cost estimates are based on national average prices. To account for potential variation in food prices across the Nation, we first scaled all product prices down to 85% of their national average and, secondly, we scaled them up to 115% and 129% of their national average consistent with the observed range of U.S. food prices [22]. The total cost of each basket was estimated at each of the three alternative price levels in addition to national average prices. 

Finally, we examined our representative household’s ability to purchase each of the 1000 baskets at all four price levels under two different assumptions about the level of SNAP benefits they receive. In June 2020, the cost of the TFP for the representative family in this study was $157.00 per week. By contrast, if current policies were in place at that time, it would have instead been 21% higher ($189.97). 

## 3. Results

### 3.1. Simulated Food Basket Characteristics

Simulation results confirm that the 1000 generated baskets are consistent in cost with the historic TFP. When evaluated at national average prices, they cost about 40% of that food plan ($63.89 of $157.00, on average) (Table 3).

The 1000 baskets created for the study also contain about 88 different products, on average (Table 3), including many items commonly eaten as standalone foods as well as others that typically serve as ingredients in recipes. Shown in Table 4 is an example basket. It is not among the 1000 baskets generated for the study; rather it was created for illustrative purposes and includes a smaller number of items for ease of exposition. However, the basket is available at the same cost level. By choosing some items costing less than $0.45 per cup-equivalent, such as fresh potatoes ($0.21 per cup-equivalent), apples ($0.43), and onions ($0.44), it is possible for our representative household to buy some higher cost items like canned corn ($0.49), canned tomatoes ($0.50), and fresh broccoli florets ($0.87), and stay within budget. The variety of items in the 1000 baskets should allow households from a variety of backgrounds to cook meals and snacks containing the meats and other types of foods they also eat. Different types of households, such as those of different racial and ethnic backgrounds, may prefer different meals. However, according to the food recipe database that USDA uses to create the TFP [29], the example basket shown in Table 4 includes all vegetables required to make shepherd’s pie (potatoes, onions, green peppers, celery, and canned tomatoes), Puerto Rican-style green pepper (green peppers, onions, fresh tomatoes), meatless spinach quiche (onions, frozen spinach), and Fajitas with chicken and vegetables (onions, red peppers, green peppers). Alternatively, a household could forgo purchasing the bag of fresh, trimmed celery in order to buy some fresh mushrooms and have all the vegetable ingredients needed to make tuna casserole with vegetables and cream (mushrooms, green peppers, onions). Much of any price differential between the celery and mushrooms could furthermore be erased by purchasing less canned fruit cocktail and more fresh apples or by purchasing less baby carrots and more fresh whole carrots. 

However, if our representative SNAP household faces higher-than-average prices, it might not be able to afford either the example basket in Table 4 or the 1000 baskets generated for this study. Below, we examine our representative household’s ability to purchase each of the 1000 baskets at 4 different price levels with maximum SNAP benefits equal to the TFP cost before and after the 2021 TFP modernization (121% of the historic TFP).

### 3.2. Affordability of the Simulated Baskets

A check of whether our representative 4-person household could purchase the 1000 food baskets generated for the study reveals how fruit and vegetable affordability can vary. The box and whiskers plot in Figure 1 shows the shares of this household’s maximum SNAP benefits that it must allocate to fruits and vegetables at each of our 4 different price levels to buy the generated baskets with benefits equal to the TFP before and after October 2021.

#### 3.2.1. Affordability by Food Price Level

Results confirm that households facing national average prices can buy a sufficient quantity and variety of fruits and vegetables for about 40% of what it would have cost them to follow the historic TFP. The least expensive of our 1000 baskets would cost our representative household 38% of that amount while the most expensive would cost it 44%. The median necessary food budget share is 41%. However, as discussed above, the average U.S. household allocates only about 26% of its food budget to fruits and vegetables [18]. The ability of a SNAP household to afford a healthy diet rich in fruits and vegetables likewise rested on that household’s willingness to prioritize these two food groups.

Households residing in communities with below average food costs had some additional flexibility even prior to the 2021 TFP modernization. Simulation results show that, if our representative 4-person households received SNAP benefits equal to the historic TFP and faced food prices 15% below national average prices, it could have afforded the baskets generated in this study with only 32% to 37% percent of its benefits.

The situation was most difficult for households facing substantially above average food prices. Prior to October 2021, a household receiving benefits equal to the previous TFP and facing food prices 15% or 29% percent higher than national average prices would have needed to allocate 44% to 50% or 49% to 56% percent of those benefits to fruits and vegetables, respectively, leaving too little money for other types of foods.

#### 3.2.2. Affordability by Thrifty Food Plan Level

Our results also illustrate how expanding the TFP in October 2021 will help SNAP households to better afford a healthy diet that meets Federal dietary recommendations for fruits and vegetables. Participating households facing all food price levels can afford the baskets generated in this study with a smaller share of their benefits than previously and therefore have less need to sacrifice spending on other types of foods. Alternatively, those who prioritize healthy eating can afford even higher cost food baskets containing a wider variety of fruits and vegetables than those generated in this study. 

Households’ situations nonetheless continue to vary. Despite helping households to better afford healthy diets, SNAP participants residing in communities with substantially above average prices must still spend about 40% of the current TFP to afford the fruits and vegetables envisioned by USDA when it created the lower cost, historic TFP. 

## 4. Discussion

Members of lower-income households have expressed the opinion that fruits and vegetables are not affordable [13,14,15,16,17]. The simulations performed in this study illustrate how the modernized TFP could help them to satisfy dietary guidance for both food groups. Results indicate that, at national average prices, the TFP cost levels set in October 2021 are sufficient to support a diet consistent with the *Dietary*
*Guidelines 2020–2025* fruit and vegetable recommendations with a budget share more in line with U.S. societal norms, less need to reduce spending on other types of foods, and more room to buy wider variety fruits and vegetables than the historic TFP envisioned.

Households’ situations nonetheless continue to vary. Not all SNAP participants face national average prices. For those in areas facing higher prices, such as large metropolitan areas, rural areas, and other communities with low access to grocery stores, meeting fruit and vegetable recommendations will still require prioritizing these two food groups to a greater extent than U.S. households generally do. This is something SNAP households may resist. Research shows that, when money is tight, households tend to prioritize meats along with grains that help to extend other foods (e.g., preparing meat combined with rice or pasta increases the number of people the meat can feed) [15,30].

While the Food and Nutrition Act of 2008 prohibits region-specific Thrifty Food Plans, with the exception of calculations for Alaska and Hawaii, this study’s results may help the administrators of SNAP-Ed programs as well as other state and local level entities to tailor their efforts around households’ unique situations. Food pantries might also target households living in high-cost areas, where the need for services may be greatest. 

Program administrators and nutrition advocates in other parts of the world besides the U.S. may similarly wish to take note of the findings of this study as variation in food prices is a global phenomenon. For example, in the European Union (EU), most countries have a free or subsidized fruit and vegetable program for children, such as the EU School Fruit Scheme [31]. Such efforts might also target households living in high-cost areas, where again the need for services may be greatest.

Several limitations of the study and our simulations should be kept in mind as we consider a family that receives maximum SNAP benefits, but abstract in important ways from SNAP households’ real-life situations. On the one hand, we assume the households do not spend any of their own money on food. We also do not account for any enhanced, COVID-19 pandemic-related benefits that SNAP households received during 2020. On the other hand, we also assume that our 4-person family has two relatively young children (aged 6–8 and 9–11 years old). In reality, some 4-person families have two teenage children and yet they still receive the same level of SNAP benefits, all else constant, despite having greater caloric needs.

USDA’s current method of adjusting SNAP benefits over time can also leave participants vulnerable to food price inflation [32]. A household’s SNAP benefits are adjusted annually in October based on its costs for following the TFP during the previous June. In other words, increases in SNAP benefits lag the food price changes for which they compensate. Indeed, since adjustments only occur once per year, nearly 16 months of price changes occur before benefits are next increased. 

Finally, among our study’s limitations, we do not account for SNAP participating households’ cooking skills and time constraints. Much research shows that these factors can limit their ability to prepare home-cooked meals rich in fruits and vegetables [33,34,35]. 

In previous years, the average SNAP household (as well as the average U.S. household generally) did not meet fruit and vegetable recommendations [4]. This gap can be filled under the new SNAP maximum benefit—although challenges will remain for some households. Still others may choose diets far from the *Dietary Guidelines* recommendations despite having the financial means to eat healthier. When the data become available, future research comparing fruit and vegetable consumption by SNAP households before and after October 2021 is needed. Such comparisons using actual consumption data will allow researchers to fully assess the impact of USDA’s 2021 TFP modernization on the healthfulness of SNAP participants’ diets, reveal whether increasing affordability increases fruit and vegetable consumption and further our understanding of inequities that continue to exist. Future research could ask if and how consumption patterns vary between higher and lower cost areas.

## 5. Conclusions 

Persons in the U.S. in general do not consume recommended quantities of fruits and vegetables. Inequities also exist. Individuals in low-income households and participating in SNAP are even less likely to meet fruit and vegetable targets than those in higher-income households. Explanations for the disparity focus on affordability. Lower-income households have tighter budgets and often do not prioritize fruits and vegetables, leaving too little money for these two food groups. While increased SNAP benefits based on the cost of the 2021 Thrifty Food Plan increase provide SNAP households with substantial additional flexibilities, households facing higher prices are likely to continue facing challenges meeting fruit and vegetable guidelines. Actual consumption changes remain to be seen. Future international efforts such as the UN’s International Year of Fruits and Vegetables may wish to consider food price variation and food assistance funding levels when considering the affordability of fruits and vegetables worldwide.

## Figures and Tables

**Figure 1 ijerph-19-02772-f001:**
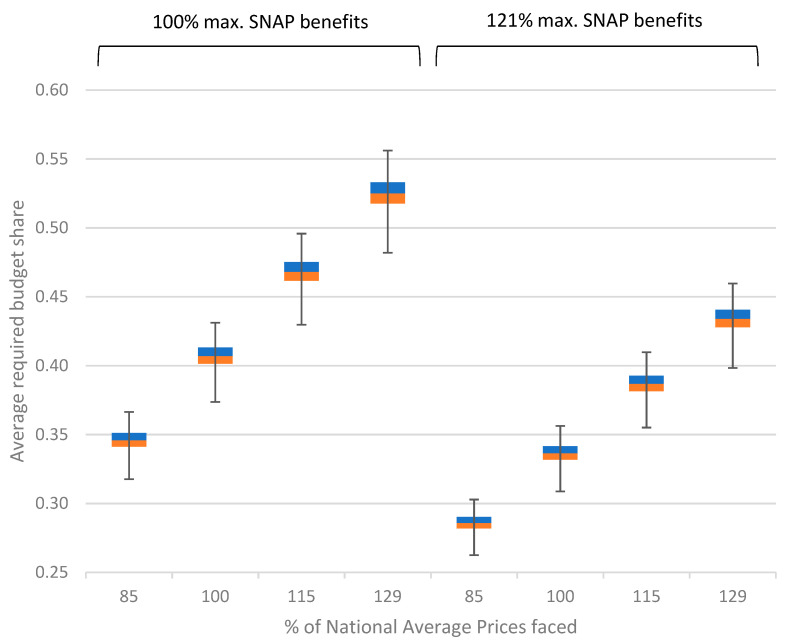
Results of simulation conducted to assess affordability of fruit and vegetable baskets sufficient to satisfy Federal dietary recommendations for both food groups for a 4-person family over 1 week with maximum SNAP benefits based on historic and modernized Thrifty Food Plan.

**Table 1 ijerph-19-02772-t001:** Costs of selected fruit per cup-equivalent, 2016 and 2020.

	2016 Cost	2020 Cost
**Whole and Cut Fruit**		
Watermelon, fresh	$0.20	$0.20
Bananas, fresh	$0.28	$0.28
Cantaloupe, fresh	$0.38	$0.38
Apples, fresh	$0.44	$0.43
Pineapple, fresh	$0.47	$0.46
Applesauce	$0.57	$0.58
Peaches, fresh	$0.60	$0.59
Raisins	$0.59	$0.60
Pears, fresh	$0.61	$0.61
Oranges, fresh	$0.66	$0.65
Honeydew melon, fresh	$0.67	$0.67
Mangoes, fresh	$0.68	$0.67
Pineapple, canned in juice	$0.69	$0.71
Plums, fresh	$0.77	$0.76
Grapes, fresh	$0.77	$0.76
Fruit cocktail, canned in juice	$0.76	$0.78
Apricots, canned in juice	$0.80	$0.81
Strawberries, fresh	$0.85	$0.85
Strawberries, frozen	$0.86	$0.88
**Juice**		
Apple, ready to drink	$0.32	$0.32
Orange, ready to drink	$0.33	$0.34
Grape, ready to drink	$0.37	$0.38
Grape, ready to drink	$0.42	$0.43
Orange, ready to drink	$0.42	$0.43
Grapefruit, ready to drink	$0.42	$0.43
Pineapple, ready to drink	$0.49	$0.50
Prune, ready to drink	$0.72	$0.74

Note: Estimates for 2020 are derived by using the Consumer Price Index to update 2016 cost estimates reported in the USDA Fruit and Vegetable Prices data product. Selected products available for less than $0.90 per cup-equivalent only. See USDA Fruit and Vegetable Prices data product for a complete list of 2016 prices.

**Table 2 ijerph-19-02772-t002:** Costs of selected vegetables per cup-equivalent, 2016 and 2020.

Dark Green Vegetables	2016 Cost	2020 Cost
Romaine lettuce, fresh head	$0.33	$0.36
Spinach, canned	$0.65	$0.67
Broccoli, cooked from frozen	$0.71	$0.72
Broccoli florets, cooked from fresh	$0.81	$0.87
Spinach, cooked from frozen	$0.85	$0.87
**Red and Orange Vegetables**		
Whole carrots, boiled from fresh	$0.30	$0.33
Baby carrots	$0.40	$0.43
Carrots, canned	$0.45	$0.46
Tomatoes, canned	$0.49	$0.50
Fresh Roma tomatoes	$0.53	$0.57
Sweet potatoes, cooked from fresh	$0.57	$0.62
Red peppers, fresh	$0.75	$0.81
Butternut squash, cooked from fresh	$0.82	$0.88
Fresh round tomatoes	$0.83	$0.89
**Beans, Peas, and Lentils**		
Pinto Beans, canned	$0.48	$0.49
Red Kidney Beans, canned	$0.51	$0.52
**Starchy Vegetables**		
Potatoes, cooked from fresh	$0.20	$0.21
Corn, canned	$0.48	$0.49
Green peas, canned	$0.54	$0.55
Corn, cooked from frozen	$0.60	$0.62
Green peas, cooked from frozen	$0.66	$0.67
**Other Vegetables**		
Green cabbage, cooked from fresh	$0.26	$0.29
Iceberg lettuce	$0.28	$0.30
Cucumbers, consumed fresh with peel	$0.34	$0.37
Green beans, canned	$0.38	$0.39
Celery, trimmed bunch	$0.40	$0.43
Onions, consumed raw	$0.41	$0.44
Green peppers, consumed raw	$0.48	$0.52
Whole mushrooms, consumed raw	$0.56	$0.61

Note: Estimates for 2020 are derived by using the Consumer Price Index to update 2016 cost estimates reported in the USDA Fruit and Vegetable Prices data product. Selected products available for less than $0.90 per cup-equivalent only. See USDA Fruit and Vegetable Prices data product for a complete list of 2016 prices.

**Table 3 ijerph-19-02772-t003:** Characteristics of 1000 simulated baskets that satisfy Federal dietary recommendations for fruits and vegetables for a 4-person family over 1 week.

Weekly Basket Characteristic	Statistic
Average cost of baskets at national average prices	$63.89
Different products in each basket (#)	88.42
Baskets costing less than $0.45 per cup equivalent (mean)	50%
Dark green vegetables (# servings ^1^)	14
Red and orange vegetables (# servings ^1^)	46
Legumes (# servings ^1^)	14
Starchy vegetables (# servings ^1^)	44
Other vegetables (# servings ^1^)	36
Fruit (# servings ^1^)	104

^1^ Servings defined as half cup equivalents.

**Table 4 ijerph-19-02772-t004:** Example of a weekly fruit and vegetable basket affordable with a 4-person family’s maximum SNAP benefits based on the Thrifty Food Plan prior to October 2021 ^1^.

Description of Product at Retail Stores	Cup-Equivalents Purchased	Estimated Cost in 2020
4 pounds of fresh apples	14.8	$6.41
4 pounds of fresh bananas	7.7	$2.18
1 bag of fresh grapes weighing 1 pound	6.0	$3.66
4 cans of fruit cocktail in juice, 15.2 ounces each	7.0	$5.48
2 half-gallon of ready-to-drink orange juice	16.0	$6.93
2 packages of frozen spinach, 10 ounces each	2.6	$2.25
1 bag of fresh broccoli florets weighing 2 pounds	5.9	$5.08
2 bags of baby carrots, 1 pound each	7.3	$3.09
1 bag of whole carrots weighing 1 pound	2.6	$0.83
2 medium-sized red peppers	3.1	$2.50
2 pounds of sweet potatoes	3.7	$2.27
2 cans of tomato, 14.5 ounces each	3.4	$1.69
6 fresh Roma tomatoes	3.6	$2.09
1 can of corn, 15.2 ounces	1.7	$0.83
2 bags of frozen green peas, 16 ounces each	5.1	$3.39
4 pounds of fresh potatoes	12.3	$2.60
1 small head of lettuce	2.9	$0.88
1 cucumber	2.5	$0.95
2 cans of green beans, 15.2 ounces each	4.1	$1.60
2 large onions	2.8	$1.24
1 bag of trimmed fresh celery	4.1	$1.80
2 medium-sized green peppers	3.0	$1.56
4 cans of pinto beans, 16 ounces each	6.7	$3.54
Total	129.0	$62.80

^1^ Household is assumed to include one adult male (31 to 45 years old), one adult female (31 to 45 years old), and two children (one aged 6–8 years old and the other aged 9–11 years old). All people are moderately active. Notes: The total cost of all fruits and vegetables in the table ($63.32) represents about 40% of the household’s overall TFP cost of foods in 2020. The total number of cup-equivalents also exceeds the recommended level by 5% to account for potential food loss due to spoilage and other factors that can lower consumption.

## Data Availability

The data presented in this study are available online at https://www.ers.usda.gov/data-products/fruit-and-vegetable-prices/ (accessed on 1 February 2022).

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
