# Peer review of "U.S. Fruit and Vegetable Affordability on the Thrifty Food Plan Depends on Purchasing Power and Safety Net Supports"

_ijerph, 2022, doi:10.3390/ijerph19052772_

Round 1

Reviewer 1 Report

The investigation made has a valuable contribution in ensuring equilibrate diet. Such studies may be considered for establishing the norms included in the Supplemental Nutrition Assistance Program. Such reports could help administrative deciders create programs in which lower-income householders are encouraged to purchase fresh fruits and vegetables based on dedicated vouchers.

The authors are recommended to insert reference(s) for the product's weight changes (pg. 3, line 135-140). Also, please include the study limitations and some recommendations and/or future perspectives.

The paper could be accepted for publication after minor changes. It has to be revised by the author(s) and resubmitted with the suggested modification specified in the reviewer's comments.

Author Response

Thank you very much for your thoughtful comments. We made the recommended changes to our paper. These include: 

  1. We inserted a reference for products’ weight changes (formerly pg. 3, line 135-140; currently page 4, lines 145).  
  2. We also expanded discussion of our study’s limitations (please see pg. 11, line 359-376). In addition to explaining how our simulation may differ from SNAP households’ real life situations, we have added some discussion of and references to studies that show households may not have the time or cooking skills necessary to prepare meals rich in fruits and vegetables. 
  3. In the final paragraph of our discussion, we also emphasize the need for future research that fully assesses the impact of USDA’s 2021 TFP modernization on the healthfulness of SNAP participants’ diets and sheds light on whether increasing affordability increases fruit and vegetable consumption.   

Reviewer 2 Report

I have carefully read the manuscript "U.S. Fruit and Vegetable Affordability on the Thrifty Food Plan Depends on Purchasing Power and Safety Net Supports". In general, the paper is well-written and well structured, but I have several remarks. I can suggest major revision, before acceptance of this paper. Some suggestions, requirements, and questions are:

  • In this kind of study, it would be desirable to emphasize some categories of consumers and the availability of different foods to certain groups of consumers.
  • Different categories of consumers have different baskets with different types of products.
  • Considering that these are only average values and st. dev., I would expect some diagram in that sense, to show some results graphically.
  • The disadvantage is that everything is described in $ and nothing in terms of some technological parameters such as mass, percentage.
  • It is shown that the entire bill was the cost price, and it is unclear what was actually obtained for that money?

I am quite indifferent to this paper, as it contains only accounting parts. I would suggest rejecting.

Author Response

Thank you for your careful read of our paper and for your comments. We have worked to answer some of your questions in the text to ensure a clarity and feel our paper is stronger thanks to your feedback. Responses to your specific points:

  • In this kind of study, it would be desirable to emphasize some categories of consumers and the availability of different foods to certain groups of consumers. 

Response: We agree that different categories of consumers might desire different baskets. Hispanic families, for example, might desire something other than a traditional American diet. The variety of vegetables and fruits costing less than 90 cents per cup-equivalent and contained in our baskets should support both. This important point is now discussed on pages 7 to 8, lines 253-266 including some discussion of the vegetables in recipes in the Food and Nutrient Database for Dietary Studies, which is one of the database that USDA uses when constructing the Thrifty Food Plan. 

  • Different categories of consumers have different baskets with different types of products. 

Response: We agree. Please see above. 

  • Considering that these are only average values and st. dev., I would expect some diagram in that sense, to show some results graphically. 

Response: We replaced table 5 with a box and whiskers plot (Figure 1). We agree that this figure better illustrates our main findings than a simple table did and thank you for this suggestion. We now discuss the distribution of baskets’ costs in addition to their mean cost. 

  • The disadvantage is that everything is described in $ and nothing in terms of some technological parameters such as mass, percentage. 

Response: All baskets contain a sufficient quantity and variety of fruits and vegetables for a 4-person household to meet recommendations for both food groups over one week (129 cup equivalents in total). We have also included an example of what fruits and vegetables one of these baskets might contain (table 4). Finally, in the revised manuscript, we now discuss some of the different meals that can be prepared using the vegetables found in the baskets (pages 7 to 8, lines 253-266). 

  • It is shown that the entire bill was the cost price, and it is unclear what was actually obtained for that money? 

Response: Please see our reply to your above comment. 

Reviewer 3 Report

Dear Authors,

I have read with great interest your manuscript on: U.S. Fruit and Vegetable Affordability on the Thrifty Food Plan Depends on Purchasing Power and Safety Net Supports

The article is important from the perspective of supporting citizens in the field of proper nutrition and prevention of civilization and diet-dependent diseases. The positive aspect of the article is the methodological and analytical part that allows predicting the effects of using the SNAP program to facilitate meeting the dietary recommendations in the U.S.

I have a few comments that I believe may add value to your paper:

  1. The introduction describes in detail the issues related to the SNAP Program and its importance for the nutrition of the poorer population in the U.S. In my opinion, it would be worthwhile to expand the theoretical introduction with an issue related to the diagnosis of the current nutritional status of the U.S. population and how the nutritional status of the beneficiaries of the SNAP program compares to this background.
  2. At the same time, I would like to explain in more detail why, from the perspective of nutrition and diet-related diseases, it is worth addressing the issue of increasing the intake of oatmeal and vegetables, what this intake looks like at the moment in different populations groups, in particular in those who may be more likely to benefit from the SNAP program.
  3. Additionally, I would refer to examples of good practices from other countries aimed at supporting people to eat well - through various support systems. Expanding the review to include these aspects will outline your study in a broader international context, which I found lacking in the paper.
  4. For the results, I think it is useful to divide them into thematic sections that will directly relate to the research questions. In my opinion, the discussion should not only refer to the U.S. perspective, but also to the solutions of other programs supporting proper nutrition in the world, which, for better or for worse, have improved the level of fruit and vegetable consumption in the poorest or selected groups of the population, e.g. U.K. healthy start.
  5. Among the specific comments I wanted to include the repeating of information in lines 44-53 and 148-157, it would be better to leave this information in the methodology.

Extending the article to include the issues raised above could, in my opinion, make it more universal and of wider interest.

Best regards,

Reviewer

Author Response

Thank you for your encouraging comments and for your careful read of our paper. Your feedback has helped us to improve the placement of our research in the literature and in the global context as well as the clarity of our results. 

1. The introduction describes in detail the issues related to the SNAP Program and its importance for the nutrition of the poorer population in the U.S. In my opinion, it would be worthwhile to expand the theoretical introduction with an issue related to the diagnosis of the current nutritional status of the U.S. population and how the nutritional status of the beneficiaries of the SNAP program compares to this background. 

Response: We have expanded our discussion of differences in fruit and vegetable intake between lower and higher income households as well as SNAP participating and nonparticipating households. 

2. At the same time, I would like to explain in more detail why, from the perspective of nutrition and diet-related diseases, it is worth addressing the issue of increasing the intake of oatmeal and vegetables, what this intake looks like at the moment in different populations groups, in particular in those who may be more likely to benefit from the SNAP program. 

Response: We have included in the Introduction mention of the potential health benefits to lower income households of eating enough fruits and vegetables - namely, improvement of diet-related outcomes which low-income households with food insecurity experience at higher rates. 

3. Additionally, I would refer to examples of good practices from other countries aimed at supporting people to eat well - through various support systems. Expanding the review to include these aspects will outline your study in a broader international context, which I found lacking in the paper. 

Response: Thank you for your suggestion to place the paper further in international context. We have added to the Introduction the international importance of the issue. Further, in both the Discussion and Conclusions we include links to international efforts to improve fruit and vegetable consumption. 

4. For the results, I think it is useful to divide them into thematic sections that will directly relate to the research questions. In my opinion, the discussion should not only refer to the U.S. perspective, but also to the solutions of other programs supporting proper nutrition in the world, which, for better or for worse, have improved the level of fruit and vegetable consumption in the poorest or selected groups of the population, e.g. U.K. healthy start. 

Response: We have added subheadings to the results section to separate results by food price level and Thrifty Food Plan / SNAP benefit level. We include implications for the international setting in both the Discussion and in the Conclusion. 

5. Among the specific comments I wanted to include the repeating of information in lines 44-53 and 148-157, it would be better to leave this information in the methodology. 

Response: Thank you for recognizing the duplication. We have removed this information from lines 44-53. 

Round 2

Reviewer 2 Report

The authors responded correctly to my comments. I have no more comments.

Author Response

Thank you again for your helpful comments that have improved this version.